# Immune Checkpoint Inhibitor Therapy in Hormone Receptor-Positive Breast Cancer

**DOI:** 10.3390/ijms262412171

**Published:** 2025-12-18

**Authors:** David Lin, Jin Sun Lee Bitar, Isabella Ma, Yuan Yuan

**Affiliations:** Cedars-Sinai Medical Center, Los Angeles, CA 90048, USA; jinsun.bitar@cshs.org (J.S.L.B.);

**Keywords:** immune-checkpoint, inhibitors, hormone, receptor-positive

## Abstract

Recent progress in immunotherapy has led to the routine use of immune checkpoint inhibitors (ICIs) in TNBC therapy and significant improvement in clinical outcomes. Incorporation of ICI into HR+/HER2− or HER2+ breast cancer has been hindered by its poor immunogenicity, and many novel combination strategies aim to convert immune cold tumors into immune hot tumors and increase the immunogenicity of HR+/HER2− breast cancer. A few recent clinical trials have shown its potential promise in high-risk HR+/HER2− early-stage breast cancer, but there is insufficient evidence to support routine use of immunotherapy in HR+ breast cancer, and longer-term follow-up is required to understand its impact on survival. This review presents an overview of immunotherapies currently under clinical development and updated key results from clinical trials, with a focus on HR+/HER2− breast cacner.

## 1. Introduction

Hormone receptor-positive/human epidermal growth factor receptor 2-negative (HR+/HER2−) breast cancer (BC) is defined by >1% expression of either estrogen receptors (ER) or progesterone receptors on the tumor cell surface. HR+/HER2− BC is the most common subtype of BC, and it is usually associated with a favorable prognosis when compared with triple-negative breast cancer (TNBC) or HER2+ BC [1]. However, 10–40% of HR+/HER2− early-stage BC (EBC) experience metastatic relapse, and HR+/HER2− subtypes such as luminal B (LumB), defined by high tumor grade and high Ki-67 proliferation marker, often have a much poorer response to standard of care treatments [2], hence a higher recurrence rate. The management of HR+/HER2− EBC is currently based on pathological stage and genomic risk factors, such as Oncotype DX and MammaPrint (MP), to optimize treatment, with high-risk patients triaged to receive adjuvant chemotherapy followed by endocrine therapy with CDK4/6 inhibitors (CDK4/6i) [3,4]. HR+/HER2− BC have generally been considered non-immunogenic and have fewer pathologic complete responses (pCR) to neoadjuvant chemotherapy (NACT). While it is evident that the HR+/HER2− subgroup exhibits less immunogenicity than TNBC, studies have shown that the LumB subtype expresses higher levels of programmed death-ligand 1 (PD-L1), tumor mutational burden (TMB), and tumor-infiltrating lymphocytes (TILs), which suggests the possibility of enhancing the current standard of care using immunotherapies such as immune checkpoint inhibitors (ICIs) [5].

Cancer immunotherapy can boost anti-tumor responses by regulating the tumor microenvironment (TME). The combination of chemotherapy and ICI pembrolizumab was recently granted FDA approval for the first-line treatment of metastatic TNBC (mTNBC) after demonstrating significant improvements in pCR, invasive disease-free survival (IDFS), and overall survival (OS) in the KEYNOTE-522 trial [6,7]. Similarly, a considerable proportion of HER2+ BC has high TILs [8,9] and high PD-L1 expression [10], and is therefore thought to have a significant benefit from immunotherapy [11], although clinical data remain mixed. In contrast, the role of ICIs remains to be further characterized in HR+/HER2− BC, which is considered “immune cold” [12]. ICIs have shown limited efficacy in the HR+/HER2− metastatic BC (MBC), likely attributed to heavily pretreated patient populations entering these trials. In contrast, cumulative evidence has shown significantly improved pCR when comparing chemo-ICI and chemotherapy alone in the neoadjuvant setting. Despite the improvements in pCR rates, the long-term efficacy of ICI in early-stage HR+/HER2− BC is eagerly awaited. At the same time, long-term toxicities of ICIs remain unknown and underscore the need to identify predictive biomarkers for better patient selection. This review paper summarizes the clinical trial development of ICI in HR+/HER2− BC. For other immunotherapeutic approaches beyond ICI, refer to our recent publication on this topic [13].

## 2. ICI in Metastatic Breast Cancer

Over the last decade, multiple strategies have been used to improve immunogenicity in HR+/HER2− BC, converting immune cold tumors into immune hot tumors. These include ICI combinations with the following: chemotherapy, endocrine therapy, targeted therapies such as histone deacetylase (HDAC) inhibitors, AKT inhibitors, CDK4/6i, and radiation therapy, as illustrated in Figure 1.

### 2.1. ICI Monotherapy (with or Without Aromatase Inhibitor) in HR+/HER2− MBC

Early clinical trials testing the efficacy of single-agent ICIs showed limited benefit in HR+/HER2− MBC. In a phase I trial combining exemestane and tremelimumab, a cytotoxic T-lymphocyte-associated protein 4 (CTLA-4)-targeting ICI, the best overall response was stable disease (SD) in 11 of 26 patients (42%), with no partial or complete objective responses observed [14]. Biomarker analysis of the study reveals an increase in inducible costimulatory-expressing T cells and a decrease in Forkhead box P3+ (FOXP3+) CD4+ regulatory T cells (Tregs), both of which likely signal an enhanced antitumor immune response [14]. KEYNOTE-028 (NCT02054806) explored the efficacy of pembrolizumab monotherapy in 25 patients with PD-L1+ (CPS ≥ 1) heavily pre-treated HR+/HER2− MBC. Median number of prior therapies for BC, including endocrine agents, was 9 (range, 3–15). A 12% overall response rate (ORR) and 1.8 months median progression-free survival (mPFS) were observed [15]. Notably, among 261 patients screened for the trial, only approximately 20% of tumors showed PD-L1 positivity in KEYNOTE-028 [15]. Similarly, the phase Ib JAVELIN trial (NCT01772004) studied avelumab monotherapy in MBC (N = 168), including 72 women with HR+/HER2− BC, with an ORR of 2.8% in patients without PD-L1 pre-selection [16]. In our previous phase II trial (NCT02648477), we found limited efficacy of pembrolizumab plus an aromatase inhibitor (AI) in a cohort of patients with HR+/HER2− MBC without PD-L1 status selection (N = 20): partial response (PR) 10%, SD 15%, and mPFS of 1.8 months [17]. Likewise, Chan et al. reported a phase II trial (NCT03393845) of pembrolizumab plus fulvestrant in HR+/HER2− MBC, also not pre-selected for PD-L1 (N = 47): 11% PR, 39% SD, mPFS of 3.2 months, and clinical benefit rate (CBR) of 36.4% at 18 weeks [18]. The above trials consistently demonstrated the limited efficacy of single-agent ICI in HR+/HER2− MBC, regardless of concurrent use of an endocrine therapy backbone.

### 2.2. ICI Combinations in HR+/HER2− MBC

ICI doublets have gained success in the treatment of solid tumors. The NIMBUS trial (NCT03789110) studied the combination of the CTLA-4 targeting ipilimumab, and the programmed cell death protein 1 (PD-1) targeting nivolumab. Patients with HER2− MBC and high TMB (≥10 mut/Mb) were selected, and the enrolled patients (N = 30) showed 10.9 mut/Mb (range: 9–11) median TMB with 20% confirmed ORR [19]. Patients with TMB ≥ 14 mut/Mb (N = 6) experienced higher response rates (60% vs. 12%; *p* = 0.041) and showed a trend towards improved mPFS and OS compared to patients with TMB < 14 mut/Mb [19]. Alternatively, the combination of durvalumab, an anti-PD-1 agent, with tremelimumab was evaluated in a pilot study (NCT03393845) among women with HER2− MBC (N = 18). The results did not show activity in patients with HR+/HER2− tumors [20].

### 2.3. Chemo-Immunotherapy Combinations in HR+/HER2− MBC

Preclinical studies demonstrated an immune-boosting effect by traditional chemotherapy: cisplatin can increase major histocompatibility complex (MHC) class I expression [21]: doxorubicin enhances antigen-specific CD8 T-Cell proliferation and tumor infiltration [22], and cyclophosphamide induces Treg depletion [23].

The combination of ICI and chemotherapy was tested in HR+/HER2− MBC. In a phase II trial (NCT03044730) combining pembrolizumab with capecitabine in HR+/HER2− MBC (N = 14), ORR 14% and CBR of 29% were reported in the biomarker-unselected, pretreated cohort [24]. Tolaney et al. reported a randomized phase II trial (NCT03051659) comparing the combination of eribulin and pembrolizumab with eribulin monotherapy (N = 88). The results showed that mPFS and ORR were similar between the two groups (mPFS, 4.1 vs. 4.2 months; hazard ratio, 0.80; 95% CI, 0.50–1.26; *p* = 0.33; ORR, 27% vs. 34%, respectively; *p* = 0.49) [25].

### 2.4. Targeted Therapy and ICI Combinations in HR+/HER2− MBC

#### 2.4.1. ICI Plus HDAC Inhibitor or AKT Inhibitor in HR+/HER2− MBC

The phase 1/2 MORPHEUS umbrella study (NCT03280563) investigated multiple atezolizumab (a PD-L1 inhibitor) combinations with targeted therapies, including entinostat, ipatasertib ± fulvestrant, in HR+/HER2− MBC chemotherapy-naïve patients in the second or third line setting [26]. The combination of entinostat, an HDAC inhibitor, with atezolizumab showed no efficacy with a mPFS of 1.8 months and ORR at 6.7% (N = 15) [27]. Atezolizumab plus fulvestrant and ipatasertib (N = 26) showed modest efficacy, with a mPFS of 4.4 months and ORR of 23.1%, with significant toxicities: rash (73.0%), diarrhea (53.8%), nausea (42.3%), fatigue (26.9%), vomiting (23.1%), hyperglycemia (11.5%) [28].

In a phase 2 trial (NCT02395627) combining tamoxifen, vorinostat, and pembrolizumab (N = 34, only 28 patients received pembrolizumab), a limited RR of 7.4% and CBR of 19% was found, and the study was terminated prematurely in unselected patients [29]. Among all five patients with clinical benefit, T-cell exhaustion (CD8+ PD-1/CTLA-4+) and treatment-induced depletion of Tregs (CD4+ FOXP3/CTLA-4+) were identified in either tumor or blood; However, only one non-responder showed similar immune markers, indicating that appropriate patient selection with the exhausted CD8+ T cell immune signature is feasible [29].

#### 2.4.2. ICI Plus CDK4/6 Inhibitors in HR+/HER2− MBC

Preclinical study conducted by Goel et al. has shown immune-modulatory effects of CDK4/6i by increasing interferon production, improving tumor antigen presentation, suppressing Treg proliferation, and increasing tumor infiltration and the activation of effector T cells, thus potentially enhancing the response to ICIs [30]. Furthermore, Deng et al. also showed CDK4/6i palbociclib and abemaciclib significantly enhance T cell activation, contributing to anti-tumor effects in vivo, due in part to de-repression of nuclear factor of activated T cell family proteins and their target genes, critical regulators of T cell function [31]. In addition, several pre-clinical tests, including a novel ex vivo organotypic tumor spheroid culture system and multiple in vivo murine syngeneic models, has shown that CDK4/6 inhibition augments the response to PD-1 blockade, thereby providing a rationale for combining CDK4/6i and ICI [31].

Our group has previously conducted a phase 1/2 trial testing the combination of pembrolizumab with palbociclib and letrozole (plus ovarian suppression for pre-menopausal women) for HR+/HER2− MBC patients (NCT02778685). Cohort 1 (N = 4) included patients who had plateaued response to palbociclib and letrozole, and a “deepened” response was observed in these patients after the addition of pembrolizumab. In cohort 2 (N = 16), patients started pembrolizumab, palbociclib, and letrozole after one cycle of “immune priming” with palbociclib and letrozole. The results showed 31% complete response (CR), 25% PR, and 31% SD, with 25.2 months (95% CI 5.3- NR) mPFS and 36.9 months (95% CI 36.9- NR) median OS in cohort 2, where the median follow-up was 24.8 months (95% CI 17.1- NR) [32]. The pretreatment specimen biomarker analysis showed overall low TILs and PD-L1 expression as well as low TMB. In the peripheral blood analysis, we found that higher pretreatment frequencies of effector memory CD45RA+ CD8+ T cells and effector memory CD4+ T cells were correlated to those that responded to palbociclib plus pembrolizumab and letrozole [33]. The same was not observed in patients treated with pembrolizumab and AI; therefore, we further characterized these effector memory T-cell subsets as baseline biomarkers of response in the combination treatment of CDK4/6i plus ICI [33]. These results provided mechanistic evidence of how CDK4/6 inhibitors prime “immune cold” tumors for improved efficacy for the combination of CDK4/6i with ICI. In the most recent update of this trial, grade 3/4 adverse events (AEs) included: neutropenia (83%), leukopenia (56%), elevated alanine aminotransferase (ALT) (13%), elevated aspartate aminotransferase (AST) (13%), and thrombocytopenia (10%), and no ILD was observed [34]. In addition, gut microbial profiling revealed multiple species that were more abundant in patients with a response (*p* < 0.05), with notable species belonging to the families Lachnospiraceae, Ruminococcaceae, and Rikenellaceae [34]. Several metabolic pathways were enriched in responders (*p* < 0.05), including pathways related to fatty acid biosynthesis and the tricarboxylic acid cycle, suggesting that ICI combination therapy in MBC may be associated with the gut microbiome [34]. These findings indicate a novel approach to biomarker analysis for ICI treatment and should be further evaluated in other studies.

The combination of abemaciclib with pembrolizumab in HR+/HER2− MBC (NCT02779751) was reported by Rugo et al. (N = 28, pretreated cohort 2) with a mPFS and mOS of 8.9 and 26.3 months [35]. Despite the demonstrated antitumor activity, the combination showed significant grade ≥ 3 AEs: neutropenia (28.6%), AST increase (17.9%), ALT increase (10.7%), diarrhea (10.7%). Furthermore, the incidence of grade ≥ 3 interstitial lung disease (ILD)/pneumonitis was 7.7% (N = 2, one each of grade 3 and grade 5); the grade 3 ILD event was later reported to lead to acute hypoxic respiratory failure after treatment discontinuation, resulting in a grade 5 event [35]. In the WJOG11418B NEWFLAME trial, nivolumab, abemaciclib plus endocrine therapy (cohort 1 fulvestrant or cohort 2 letrozole as endocrine backbone) was tested in a total of 16 patients, with ORR 50%, and the most common grade ≥ 3 treatment-related AEs (TRAEs) was neutropenia (7 (58.3%) and 3 (60.0%) in the fulvestrant and letrozole cohorts, respectively), followed by alanine aminotransferase elevation (5 (41.6%) and 4 (80.0%)). The study was terminated due to a treatment-related death that occurred due to ILD [36]. In both studies, the concern of increased toxicities when combining abemaciclib with ICI has hindered its further development.

On the other hand, the combination of atezolizumab with abemaciclib and fulvestrant was tested in the Phase I/II MORPHEUS breast cancer study (NCT03280563). The reported mPFS was 6.3 months in the atezolizumab–abemaciclib–fulvestrant arm, vs. 3.2 months in the abemaciclib–fulvestrant arm [37]. In contrast to the two previous trials, only grade 1/2 ILD/pneumonitis (7.7%) was observed in the triplet arm [37]. Moreover, the study initiated an investigation on the combination of atezolizumab, abemaciclib, and an oral selective ER downregulator giredestrant in HR+/HER2− MBC. The initial results were recently reported at ASCO2025: seven of nine patients had confirmed responses in patients with *ESR1*-mutated disease [38]. The full study result is eagerly awaited for this ongoing clinical trial.

#### 2.4.3. ICI Plus PARP Inhibitors in HR+/HER2− MBC

Poly (ADP-ribose) polymerase inhibitors (PARPi) combined with immunotherapy have shown antitumor activity in preclinical studies. The study results suggest that PARPis may enhance the immune response to tumor cells not only by activating an innate response through the release of tumor neoantigens and danger-associated molecular patterns, but also by stimulating an adaptive immune response and by reshaping the TME [39]. The combination of PARPi olaparib with durvalumab in patients with germline *BRCA*-mutated MBC (MEDIOLA trial, NCT02734004) showed an encouraging mPFS of 9.9 months and mOS of 22.4 months in the HR+/HER2− MBC subset (N = 16) [40]. Similarly, in the JAVELIN PARP Medley trial (NCT03330405), ICI avelumab plus talazoparib showed an ORR of 34.8% in HR+/HER2− DNA damage response-positive (DDR+) BC (N = 23) [41]. Durable responses were observed, with a promising mPFS of 5.3 months and duration of response (DOR) of 15.7 months [41]. Overall, PARPi treatment in *BRCA*-mutant or DDR+ tumors could activate the STING pathway, hence leading to enhanced immune activation [42]. Immune response stimulation with PARPis should be further tested in larger phase III trials.

### 2.5. ICI Plus Radiation Therapy in HR+/HER2− MBC

Radiation therapy is commonly used for palliation in patients with HR+/HER2− MBC. Radiation therapy is viewed as an immunotherapy based on the abscopal effects in clinical observation [43]. Furthermore, multiple pre-clinical studies have shown that radiation therapy can induce an immune response through inducing DNA damage, cytokine release, and the release of tumor neoantigens, which can result in immunogenic cell death, priming and activation of cytotoxic T cells, and T-cell infiltration into TME. The combination of ICIs with radiation therapy has demonstrated the safety and potential efficacy in BC, with ongoing trials aiming to optimize the treatment regimen with novel ICI or targeted therapy combinations and identify predictive biomarkers for better patient selection.

A phase II clinical trial (NCT02730130) combining pembrolizumab with hypo-fractionated radiation therapy (5x6Gy) in mTNBC reported an ORR of 17.6%, and three of the nine PD-L1+ tumors achieved complete and durable responses [44]. ICI plus radiation therapy combination trials have been less successful to date in HR+/HER2− MBC [45]. No clinical responses were seen in a trial (NCT03366844) by Barroso-Sousa et al., where eight HR+/HER2− patients were treated with pembrolizumab plus radiation therapy (5x4Gy) to ≥1 metastatic lesion [46]. Similarly, in the KBCRN-B002 trial (NCT03430479), which included 20/28 HR+/HER2− patients, nivolumab plus radiation therapy (1x8Gy) to a bone metastasis only had an ORR of 7% [47].

In Table 1, we provided a comprehensive summary of all the above immunotherapy trials in HR+ MBC. Noted, many of these studies are phase I or II, single-arm in nature, hence the studies are limited by small sample size and lack of control, leading to a lack of adequate statistical power and a higher likelihood of being biased. Other factors limiting these studies include heavily pretreated patient cohorts and a lack of appropriate biomarker selection, which are explored by the later neoadjuvant setting studies that will be discussed below.

## 3. Neoadjuvant ICI in HR+/HER2− Early-Stage Breast Cancer

In EBC, neoadjuvant systemic therapy provides an ideal opportunity to downstage, to assess response to therapy, and to determine the need for escalation or de-escalation of systemic therapy [48]. Across all breast cancer subtypes, pCR has been proven to be prognostic—patients who achieve pCR after NACT have a lower risk of recurrence and better OS when compared to patients who do not achieve pCR [49]. However, pCR rates are generally low among patients with HR+/HER2− EBC who are treated with conventional NACT: approximately 5–15% only [48,50]. To enhance the pCR rate, ICIs, in combination with chemotherapy or targeted therapy, are being increasingly investigated in the neoadjuvant setting. As summarized in Table 2, a variety of neoadjuvant trials incorporating immune checkpoint inhibitors were conducted and will be discussed in detail as following.

### 3.1. Neoadjuvant CDK4/6i and Nivolumab in HR+/HER2− EBC

Neoadjuvant treatment with nivolumab, palbociclib, and anastrozole in patients with HR+/HER2− breast cancer was evaluated by the phase 1b/2 CheckMate 7A8 study (NCT04075604). Among the first 21 patients treated (palbociclib 125 mg group: N = 9; palbociclib 100 mg group: N = 12), dosing-limiting toxicities were reported in 2 (22.2%) patients in the palbociclib 125 mg group [51]. Later a total of nine patients discontinued treatment due to toxicity: grade 3/4 hepatic adverse events (N = 6), grade 3 febrile neutropenia (N = 1), grade 1 pneumonitis (N = 1), and grade 3 rash and grade 2 immune-mediated pneumonitis (N = 1) [51]. The high incidence of grade 3/4 hepatotoxicity and treatment discontinuation suggests terminating further investigation of this combination treatment.

### 3.2. Neoadjuvant ISPY2 Trial in HR+/HER2− EBC

Several earlier studies built upon the knowledge that primary HR+/HER2− EBC is often less immunosuppressed than heavily pretreated MBC, and that cytotoxic agents may elicit antitumor immune responses when combined with ICI. I-SPY2 is a unique ongoing multicenter, multicohort, dynamic platform trial (NCT01042379), designed to evaluate the efficacy of novel agents in combination with NACT backbone for high-risk EBC, and these novel agents are “graduated” if they meet a predefined, subtype-specific efficacy threshold [52]. For high-risk HR+/HER2− EBC, eligibility criteria to enter the I-SPY study include a tumor size of at least 2.5cm, as well as MP high-risk.

I-SPY2 trial testing neoadjuvant pembrolizumab (q3w×4) with concurrent weekly paclitaxel followed by dose-dense doxorubicin plus cyclophosphamide (ddAC) (×4), showed a promising estimated pCR of 30% vs. 13% in the chemotherapy alone arm (N = 40), which led to the graduation of pembrolizumab from I-SPY2 [53]. The study also reported a similar 3-year event-free survival (EFS) between the two groups, and that TRAEs were mostly grade 1–2, although it was noted that six patients developed adrenal insufficiency [53]. In the HR+/HER2− EBC subgroup, whole transcriptome microarray of the pre-treatment baseline biopsies and a signature of 53 genes (ImPrint) were utilized to predict pCR to pembrolizumab with >80% and 85% of sensitivity and specificity, respectively, with a positive predictive value of 77% for the HR+/HER2− subgroup [54].

### 3.3. Neoadjuvant GIADA Trial in HR+/HER2− EBC

The phase II GIADA trial (NCT04659551) enrolled patients with stage II-IIIA, LumB-like breast cancer (defined by HR+/HER2−, Ki67 ≥ 20%, and/or histologic grade 3). All patients received epirubicin plus cyclophosphamide (EC) with concomitant triptorelin, followed by nivolumab with concurrent exemestane [55]. High rates of liver toxicity during nivolumab therapy were observed: Grade 3 ALT and AST increase occurred in 10% and 17% of patients, respectively, and grade 3 GGT increase occurred in 17% of patients. Despite being comparatively favorable to other similar studies at the time, GIADA II only showed modest results: a pCR rate of 16.3% and a residual cancer burden (RCB) 0–1 rate of 25.6% [55].

### 3.4. Neoadjuvant KEYNOTE-756 Trial in HR+/HER2− EBC

Following the initial success of the ISPY-2 trial, the phase III KEYNOTE-756 (KN-756) trial (NCT03725059) tested the role of neoadjuvant pembrolizumab in stage II-II HR+/HER2− EBC. A total of 1278 patients with HR+/HER2− high-risk EBC (T1c-2 (≥2 cm) and N1–2 or T3–4 and N0–2, grade 3) were randomly assigned (1:1) to neoadjuvant pembrolizumab or placebo with concomitant paclitaxel for 12 weeks, followed by AC or EC (q2w/q3w×4). The study continues post-operatively, where patients received adjuvant pembrolizumab or placebo plus adjuvant endocrine therapy for nine more cycles. At the study’s prespecified first interim analysis, the pCR was 24.3% (95% CI, 21.0–27.8%) in the pembrolizumab–chemo arm and 15.6% (95% CI, 12.8–18.6%) in the placebo–chemo arm (the estimated difference was 8.5 percentage points; 95% CI, 4.2–12.8; *p* = 0.00005) [56]. The pembrolizumab–chemo arm had more grade 3/4 TRAE at 52.5% in comparison to 46.4% of the placebo–chemo arm, and grade 3/4 immune-mediated AEs were reported in 45 (7.1%) and 8 patients (1.2%), respectively [56]. Notably, the most frequently occurring immune-mediated AEs were hypothyroidism (17.5% and 1.7%, respectively), hyperthyroidism (9.0% and 0.5%, respectively), and pneumonitis (2.8% and 1.4%, respectively) [56]. The study’s other co-primary endpoint of EFS is not yet mature and remains to be evaluated.

### 3.5. Neoadjuvant CheckMate 7FL Trial in HR+/HER2− EBC

Similarly, the phase 3 CheckMate 7FL trial (NCT04109066) randomized patients with HR+/HER2− EBC (T1c-2 and N1-2 or T3-4 and N0-2, grades 2–3) to receive neoadjuvant nivolumab or placebo (q3w) in combination with paclitaxel (qw) for 12 weeks, followed by AC. Post-operatively, nivolumab or placebo is continued with endocrine therapy for up to seven cycles. The results indicated a significantly increased pCR of 24.5% for the nivolumab–chemo arm, in comparison to the placebo–chemo arm at 13.8% [57]. In addition, more pronounced improvements were seen in the stage III subgroup with nivolumab arm pCR at 30.7% to 8.1% placebo, as well as the PD-L1+ subgroup at 44.3% pCR to 20.2% placebo [57]. Nevertheless, patients enrolled in the nivolumab–chemo arm experienced higher rates of grade 3–4 adverse events and treatment discontinuations when compared with those in the placebo–chemo arm.

Currently, the IDFS and OS data for both KN-756 and CheckMate 7FL are not yet available. Longer-term follow-up for these endpoints is critical for future implementation of these therapies in routine clinical practice.

### 3.6. Neoadjuvant Radiation and ICI Trial in HR+/HER2− EBC

Radiation therapy induces immunogenic cell death and increases the release of tumor-specific antigens that recruit immune cells to the TME and upregulate PD-L1 expression [58,59]. The combination with pembrolizumab in pretreated mTNBC, regardless of PD-L1 status, showed promising efficacy [44]. Based on the hypothesis that the window regimen of preoperative pembrolizumab and radiation therapy to the primary tumor could enhance responses to NACT without substantive delay, the PEARL phase Ib/II trial (NCT03366844) was designed to assess the immune-activating effect of pembrolizumab (×2) with radiation therapy (3x8Gy) in patients with breast cancer in whom NACT was the standard of care [60]. A total of 12 HR+/HER2− EBC patients were included. A pCR of 33.3% was achieved, and 41.6% had a near pCR/RCB 0–1) [60]. Although the number of patients is small, this study highlighted the promise of radiation in conjunction with immunotherapy as a potential tool for boosting immune response in HR+/HER2− EBC. In the Neo-CheckRay trial’s (NCT03875573) safety run-in, six LumB EBC patients received concurrent neoadjuvant stereotactic body radiation therapy (SBRT) (3x8Gy), durvalumab, and oleclumab (an anti-CD73 antibody) [61]. Two patients achieved pCR, while another two patients only had scattered residual disease left [61]. After the safety run-in, the randomized phase 2 trial included the following three arms (N = 45 each): arm 1: neoadjuvant paclitaxel (qw × 12) with SBRT at week 5 (3x8Gy targeting the primary tumor only), followed by dose-dense EC; arm 2: arm 1 plus durvalumab; and arm 3: arm 2 plus oleclumab [62,63]. The pathological responses for the chemo–SBRT, chemo–Durvalumab–SBRT, and chemo–Durvalumab–Oleclumab–SBRT arms were: pCR/RCB 0 (17.8%, 31.8% and 35.6%); RCB 0–1 (37.8%, 50%, 51.1%) [64]. Grade 3/4 TRAEs were: 27%, 64.7% and 70.8%, respectively, with similar serious AEs (10.4%, 17.6% and 16.7%, respectively) [64]. This exciting clinical trial result demonstrated that higher pCR can be achieved with immuno-radiation boost, and longer-term follow-up with additional outcomes, such as IDFS, is required for a full understanding of the benefits of such therapy. Moving forward, the nuances of RT dose, fractionation, specific type of immunotherapy combination, and sequencing, coupled with the diverse biology of breast cancers and uncertainties about their impact on responses to radioimmunotherapy, underscore the need for further exploration of such an approach [65]. An ongoing METEORITE clinical trial combines neoadjuvant pembrolizumab, RT, and neoadjuvant myeloid cell suppression in patients with early-stage breast cancer, including TNBC, with the plan of adding HR+/HER2− high-risk EBC (NCT05491226).

### 3.7. Other Ongoing Neoadjuvant ICI Trials in HR+/HER2− EBC

Utilizing pCR as a reliable representative endpoint for long-term outcomes has yet to be verified in the HR+ population [66]. Despite the improved pCR seen in both KN756 and CheckMate 7FL, longer-term efficacy measurements, such as EFS and OS, will ultimately be critical in guiding the incorporation of this strategy into routine clinical practice. The ongoing SWOG S2206 trial (NCT06058377) aims to study the benefit of neoadjuvant Durvalumab, another PD-L1 inhibitor, in combination with NACT for the HR+/HER2− high-risk (MP-High2) EBC patient population [67].

**Table 2 ijms-26-12171-t002:** Summary of ICI trials in HR+/HER2− Neoadjuvant Setting.

Trial Name	Phase	Eligible Stages	Treatment Arms	Cohort	pCR	Additional Outcomes	Safety and Toxicities	Biomarker	NCT	Ref.
CheckMate 7A8	2	cT1c-T3, ≥2 cm tumor	Nivolumab + palbociclib (125 mg: 100 mg) + anastrozole	9:12	0%:8%	0%:8% RCB 0–1	89%:75% G3/4 AEs, and a total of nine pts discontinued due to AEs	PD-L1	NCT04075604	[51]
I-SPY2	2	T2-T3, ≥2.5 cm tumor (clinical) or ≥2 cm tumor (imaging)	Paclitaxel ± pembrolizumab → ddAC (P + P:P)	40:96	30%:13%	46%:24% RCB 0–1	29:33 G3/4 AEs were reported (includes TNBC pts, N = 69: 181) *	MP	NCT01042379	[53]
GIADA	2	T2-3a, LumB-like BC (Ki67 ≥ 20% and/or histologic G3)	EC + triptorelin → nivolumab + exemestane	43	16%	26% RCB 0–1	30 G3/4 AEs were reported, and nine pts discontinued due to AEs	Breast Cancer 360TM TILs	NCT04659551	[55]
KEYNOTE 756	3	cT1c-2 (≥2 cm) N1-2 or cT3–4 N0-2; G3	Paclitaxel ± pembrolizumab → ddAC (P + P:P)	635:643	24%:16%	35%:24% RCB 0–1	53%:46% G3+ TRAEs, 121: 65 pts discontinued, and one death in P + P arm due to TRAEs	PD-L1	NCT03725059	[56]
CheckMate 7FL	3	2–5 cm cN1-2 or cT3-4 cN0-2; G3 or G2 if ER 1-10%	Paclitaxel ± nivolumab → ddAC ± nivolumab (P + N:P)	257:253	25%:14%	31%:21% RCB 0–1 EFS rate 89%:92%	35%:32% G3/4 TRAEs, 30: seven pts discontinued due to AEs; in P + N arm, there are three G5 treatment-unrelated AEs, and two deaths due to TRAEs	PD-L1 TILs	NCT04109066	[57]
PEARL	1/2b	cT1-3 (≥2 cm); High risk (2 of: G2-3, Ki67 ≥ 20%, ER < 75%)	Pembrolizumab → RT → NACT (regimen decided by treating physician)	12	33%	42% RCB 0–1	41% G3/4 AEs, and 14% G3 TRAEs	PD-L1 TILs	NCT03366844	[60]
Neo-CheckRay (2023)	1b	cT2-3 N0 or T1b-3 N1-3, Ki67 ≥ 15% or G3; MP high-risk	Paclitaxel + durvalumab + oleclumab + RT → ddAC + durvalumab + oleclumab	6	33%	66% RCB 0–1	17% G3/4 AEs	TILs, PD-L1, CD73 and MHC-I	NCT03875573	[61]
Neo-CheckRay (2025)	2	cT2-3 N0 or T1b-3 N1-3, Ki67 ≥ 15% or G3; MP high-risk	Paclitaxel ± durvalumab ± oleclumab + RT → ddAC + durvalumab + oleclumab (P + D + O:P + D:P)	45:45 **:45	36%:30% **:18%	51%:49% **:38% RCB 0–1	71%:65%:27% G3/4 TRAEs	TILs, PD-L1, CD73 and BMI	NCT03875573	[62,64]
SWOG S2206	3	cT2-3 and MP-High2	AC-T ± durvalumab (A + D:A)	accruing	NA	NA	NA	PD-1, PD-L1	NCT06058377	[67]

* = Safety cohort includes TNBC (and HER2+) patients in the study, ** = Arm2 results with N = 43, two more surgeries pending at time of abstract.

## 4. Biomarkers Predicting Response to ICI in HR+/HER2− EBC

The immunology of HR+/HER2− BC is complex, characterized by low TIL infiltration (≤10% in stromal tissue), human leukocyte antigen class I downregulation, and increased lymphocyte activation gene 3 (LAG3) and T cell immunoglobulin and mucin domain 3 expression, leading to poor recruitment of activated immune cells and increased tumor-associated macrophages (TAMs) [12]. In particular, TAMs are highly immunosuppressive and are associated with tumor progression and treatment (chemotherapy and ICI) resistance [12].

Several biomarkers indicating immune activation have been associated with better response to ICI–chemotherapy combination therapy in BC: higher TILS, higher TMB, basal molecular subtype, and immune-related gene-expression signatures. TILs quantification has been associated with response to NACT in HR+/HER2− EBC patients, with pCR rates of approximately 15% in tumors with TILs > 10% [68]. In the abovementioned phase II GIADA trial, patients with TILs ≥ 10 had a 71% (5/7) pCR rate, which is significantly higher than the 16.3% overall pCR rate [55]. Of note, the PAM50 Basal BC subtype, characterized by increased TILs, showed a 50% pCR rate that is significantly higher compared with other subtypes (Luminal A 9.1%; LumB 8.3%) [55]. The GIADA trial showed that TILs, immune-related gene-expression signatures, and specific immune cell subpopulations by multiplex immunofluorescence are all associated with increased pCR [55].

Both the KN756 and CheckMate 7FL studies demonstrated ICI benefit regardless of disease stage and nodal involvement. However, the most significant ICI benefit is correlated with the ER-low patient population—defined as ER staining 1–9% in KN756, or ER staining < 50% or PR staining ≤ 10% in CheckMate 7FL. The ER-low/HER2− BC behaves similarly to TNBC in developing molecularly heterogeneous, non-luminal TMEs. In the KN756 trial, 6% of the participants enrolled in the pembrolizumab arm were ER-low, and this population demonstrated the biggest pCR rate difference (25.6%). ER-low and PD-L1+ patients showed a pCR rate difference of 24.3% in contrast to ER > 10% and PD-L1+ patients at 9.2%; Similarly, TILs ≥ 30% correlates with better EFS outcome for ER-low BC, but not for ER > 10% BC [69]. Overall, TILs and PD-L1 more effectively predict pCR in low-ER BC, but these prognostic markers showed little predictive outcome for the rest of the ER+ population, indicating that new predictive biomarkers need to be identified for the HR+/HER2− subtype.

Huppert et al. recently reported a combined analysis of pCR and distant recurrence-free survival (DRFS) rates for HR+/HER2− patients in eight neoadjuvant arms of the I-SPY2 clinical trial (pembrolizumab, N = 40; ganetespib, N = 48; ganitumab, N = 58; MK2206, N = 28; trebananib, N = 62; veliparib/carbo, N = 32; neratinib, N = 17; paclitaxel, N = 94). The analysis covers the following clinical/molecular features: patient age, disease stage, histology type, ER percentage, ER/progesterone receptor status, MP-High1 vs. MP-High2 status, BluePrint (BP)-Luminal-type versus BP-Basal-type, and ImPrint immune signature [48]. pCR rates were higher in patients with ductal versus lobular histology (pCR 19% vs. 11%, *p* = 0.049), stage II versus III disease (pCR 21% vs. 9%, *p* = 0.0013), ER percentage ≤ 66% vs. > 66% (pCR 35% vs. 9%, *p* = 3.4 × 10^−9^), MP-High2 versus MP-High1 disease (pCR 31% vs. 11%, *p* = 1.1 × 10^−5^), BP Basal-type versus Luminal-type disease (pCR 34% vs. 10%, *p* = 1.62 × 10^−7^), and ImPrint+ vs. ImPrint− disease (pCR 38% vs. 10%, *p* = 1.64 × 10^−9^) [48]. Remarkably, a subset of patients with high molecular risk was more likely to achieve pCR compared to those with less molecular risk if they exhibited overlapping MP-High2, BP-Basal-type, and ImPrint+ signatures [48].

In a separate analysis by Wolf et al., a total of 204 HR+/HER2− (MP high-risk) patients from five ICI treatment arms (anti-PD-1, anti-PD-L1 plus PARPi, anti-PD-1 plus toll-like receptor 9 agonist, and anti-PD-1 ± LAG3 antibody) and 191 patients from the chemotherapy-only control arms were included [70]. The overall pCR rate across the five ICI treatment arms is 33%. Although only 26% of HR+/HER2– patients were ImPrint+, pCR rates with ICI treatment were 75% in ImPrint+ versus 17% in ImPrint−, with the highest pCR rate > 90% in the anti-PD-1—LAG3 antibody arm; in contrast, pCR rates were 33% in ImPrint+ and 8% in ImPrint− for the chemotherapy-only control arms [70]. Tumor grade 3, MP-High2, and ER-low showed higher pCR rates of 45%, 56%, and 63%, respectively, in the ICI treatment arms [70]. This analysis highlighted that an accurate biomarker selection strategy for HR+/HER2− patients could effectively achieve pCR rates comparable to what is seen from the best neoadjuvant therapies in TNBC and HER2+ (i.e., pCR rate > 65–70%). I-SPY2.2 is now incorporating these biomarker selection strategies to optimize treatment selection for specific patient populations. For example, the ongoing SWOG S2206 only selects for patients with MP-High2 gene signatures.

Despite encouraging correlative data in TILs, ER-low, and ImPrint, none of these are routinely available in a standardized pathology report. TILs reproducibility across different pathologists remains challenging, and there has been tremendous effort in standardizing the TILs reading through the International TILs working group [71]. Similarly, ER-low signatures should be standardized. The KN-756 and CheckMate 7FL studies both showed improved benefit of ICIs for the ER-low population, yet both studies had defined the ER-low population differently. ImPrint has been used and validated through the ISPY-2 trial, but is not yet widely available outside of a clinical trial setting. Anticipating ICI approval in the HR+/HER2− neoadjuvant setting, an FDA-required companion diagnostic test will be critical to ensure that the appropriate patient is selected. Equally important, standardization of biomarker analysis is essential to enable reliable and reproducible clinical application.

Immune checkpoint inhibitor is not yet a standard of care neoadjuvant treatment option. If it becomes FDA-approved in the future, clinical application of such a treatment modality needs to be carefully weighed against its non-trivial toxicities of ICIs. Considering the ever-expanding treatment options of endocrine therapy with adjuvant CDK4/6i and or ER targeted therapy, it is critical to select the right patient for chemoimmunotherapy combination with specific biomarkers such as RNA-based molecular subtyping, high-risk by MP, high stromal TILs infiltrate, and high expression of immune signatures.

## 5. Conclusions

The development of novel immunotherapies for HR+/HER2− breast cancer is incredibly challenging. While studies in the heavily pretreated metastatic setting continue to show modest efficacy, emerging phase III data of ICI chemotherapy combinations in the neoadjuvant setting show significantly improved pCR rates, which indicates practice changing potentials of the combination. However, long-term follow-up data is required to understand its impact on patient survival. Meanwhile, the risk of immune checkpoint inhibitor-related long-term toxicities are not trivial and must also be carefully considered when incorporating these therapies into clinical practice.

## Figures and Tables

**Figure 1 ijms-26-12171-f001:**
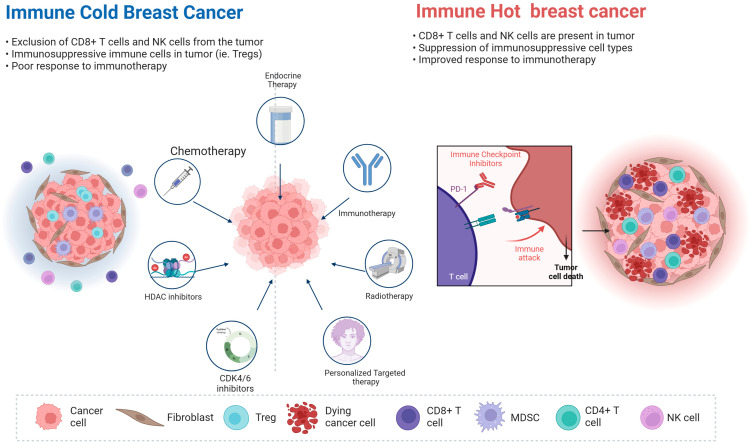
Therapies combined with checkpoint inhibition in the treatment of HR+/HER2− breast cancer: immune cold tumors can be converted into immune hot tumors through combination therapies, which activate CD8+ T cells, NK cells, and suppress Treg and elicit effective tumor killing. Credit: https://www.biorender.com/.

**Table 1 ijms-26-12171-t001:** Summary of ICI trials in HR+/HER2− MBC Trials.

Trial Name	Phase	Prior Treatment	Treatment Arms	Cohort	ORR	Survival Outcomes	Additional Outcomes	Safety and Toxicities	Biomarker	NCT	Ref.
2.1 ICI Monotherapy											
KEYNOTE-028 (2018)	1b	Pretreated with ≥1 line of therapy	Pembrolizumab	25	12%	mPFS 1.6 mo mOS 8.6 mo	CBR 20% DOR 12.0 mo	16% G3/4 AEs, and five pts discontinued due to AEs	PD-L1	NCT02054806	[15]
JAVELIN Solid Tumor (2018)	1b	Pretreated with 1–3 lines of therapy	Avelumab	72	3%	mPFS 6.0 weeks mOS 9.2 mo	DCR 28%	14% G3/4 TRAEs, eight pts discontinued, and two deaths due to TRAEs (all MBC pts, N = 168) *	PD-L1	NCT01772004	[16]
ICI Monotherapy with endocrine therapy											
Vonderheide, R.H. et al. (2010)	1	Pretreated with ≥1 line of therapy	Tremelimumab + exemestane	26	No PR/CR	NA	42% with SD ≥ 12 weeks	27% G3 AEs, and one pt discontinued due to TRAEs	Circulating CD4+ and CD8+ T cells	NA	[14]
Ge, X. et al. (2022)	2	First line or pretreated with 1–7 lines of therapy	Pembrolizumab + anastrozole/letrozole/exemestane	20	10%	mPFS 1.8 mo mOS 17.2 mo	CBR 20%	10% G3 AEs	TIL, PD-L1	NCT02648477	[17]
Chan, N. et al. (2025)	2	Pretreated with ≤2 lines of therapy	Pembrolizumab + fulvestrant	47	NR	mPFS 3.2 mo	CBR 36% mDOR 6 mo	15% G3 TRAEs, and one pt discontinued due to TRAEs	PD-L1	NCT03393845	[18]
2.2 ICI Combinations											
Santa-Maria, C.A. et al. (2018)	2	Pretreated with ≥1 line of therapy	Durvalumab + tremelimumab	11	0%	NA	CBR 19%	17 G3 AEs (includes TNBC pts, N = 18) *	none	NCT03393845	[20]
NIMBUS (2025)	2	First line or pretreated with 1–3 lines of therapy (1 + ET)	Ipilimumab + nivolumab	21	14%	mPFS 1.4 mo mOS 16.9 mo	NR	27% G3 AEs, and two pts discontinued due to AEs (includes TNBC pts, N = 30) *	TMB	NCT03789110	[19]
2.3 Chemo-immunotherapy Combinations											
Shah, A.N. et al. (2020)	2	Pretreated with ≥1 line of ET	Pembrolizumab + capecitabine	14	14%	mPFS 5.1 mo	CBR 28%	G3+ AE in 10+% pts one death from TRAE (includes TNBC pts, N = 16) *	PD-L1	NCT03044730	[24]
Tolaney, S.M. et al. (2020)	2	Pretreated with ≥2 lines of ET and ≤3 chemotherapies	Eribulin ± pembrolizumab (E + P: E alone)	44:44	27%:34%	mPFS 4.1:4.2 mo	CBR 48%:50%	68%:61% G3/4 AEs, and two deaths due to TRAEs in the E + P arm	PD-L1 TMB TILs	NCT03051659	[25]
2.4 Targeted therapy and ICI Combinations HDAC inhibitors											
Terranova-Barberio, M. et al. (2020)	2	Pretreated with 1 + lines of therapy	Pembrolizumab + vorinostat + tamoxifen	34	7%	NR	CBR 19%	33% G3/4 TRAEs, and one pt discontinued due to TRAE	PD-L1 TILs	NCT02395627	[29]
MORPHEUS (2021)	1b/2	Pre-treated with 2/3 lines of therapy, but CT-naive	Atezolizumab + entinostat: fulvestrant alone (A + E: F)	15:14	7%:0%	mPFS 1.8:1.8 mo	NR	40%:21% G3/4 AEs, and one pt discontinued due to TRAE in the A + E arm	PD-L1	NCT03280563	[27]
AKT inhibitors											
MORPHEUS (2022)	1b/2	Pre-treated with 2/3 lines of therapy, but CT-naive	Atezolizumab + ipatasertib + fulvestrant (A + I + F:F)	26:15	23%:0%	mPFS 4.4:1.9 mo	NR	62%:20% G3/4 AEs, two pts discontinued due to TRAEs in the A + I + F arm	PD-L1 PI3K status CD8-panCK	NCT03280563	[28]
CDK4/6 inhibitors											
Yuan, Y. et al. (2021) Cohort 1	½	Pretreated with palbociclib + letrozole for 6+ mo	Pembrolizumab + palbociclib + letrozole	4	50%	NR	CBR 100%	5 G3/4 TRAEs, and one pt discontinued due to TRAEs	TILs PD-L1	NCT02778685	[32]
Yuan, Y., et al. (2021) Cohort 2	2	NR	Pembrolizumab + palbociclib + letrozole	16	56%	mPFS 25.2 mo mOS 36.9 mo	CBR 88%	30 G3/4 TRAEs, and three pts discontinued due to TRAEs	TILs PD-L1	NCT02778685	[32]
Rugo, H.S. et al. (2020) Cohort 1	1b	First line	Pembrolizumab + abemaciclib + anastrozole	26	23%	NR	CBR 39% DCR 85%	69% G3+ TRAEs, nine pts discontinued, and two deaths due to TRAEs	NR	NCT02779751	[35]
Rugo, H.S. et al. (2020) Cohort 2	1b	Pretreated with 1–2 lines of CT	Pembrolizumab + abemaciclib	28	29%	mPFS = 8.9 mo mOS = 26.3 mo	CBR 46% DCR 82%	54% G3+ TRAEs, six pts discontinued, and one death due to TRAEs	NR	NCT02779751	[35]
WJOG11418B NEWFLAME Ful Cohort (2023)	2	First line or pretreated with 1+ lines of ET	Nivolumab + abemaciclib + fulvestrant	12	55%	NR	DCR 91%	92% G3+ TRAE, and seven pts discontinued due to AEs	PD-L1	NR	[36]
WJOG11418B NEWFLAME Let Cohort (2023)	2	First line	Nivolumab + abemaciclib + letrozole	5	40%	NR	DCR 80%	100% G3+ TRAE, three pts discontinued, and one death due to AEs	PD-L1	NR	[36]
MORPHEUS (2024)		Pretreated with 2/3 lines of therapy, but CT-naive	Fulvestrant ± atezolizumab ± abemaciclib (F + At + Ab:F + At:F)	38:30:20	26%:10%:10%	mPFS 6.3:3.2:2.0 mo	CBR 74%:43%:15%	82%:27%:15% G3/4 AEs, and 8:2:0 pts discontinued due to AEs	PD-L1, CD8	NCT03280563	[37]
PARP inhibitors											
MEDIOLA (2020)	1b/2	Pretreated with 1+ ET and ≤2 CT (must include taxane or anthracycline)	Durvalumab + Olaparib	16	NR	mPFS 9.9 mo mOS 22.4 mo	92% DCR at 12 weeks	33% AEs, and three pts discontinued due to AEs (including TNBC pts, N = 34) *	PD-L1, TMB	NCT02734004	[40]
JAVELIN PARP Medley (2023)	2	Pretreated with 1+ lines of CT	Avelumab + talazoparib	23	35%	mPFS 5.3 mo	DOR 15.7 mo	57% G3/4 AEs (includes data from all cohorts); one pt discontinued due to TRAE in HR + cohort	PD-L1, TMB, CD8, BRCA	NCT03330405	[41]
2.5 Radiation and ICI Combinations											
Barroso-Sousa et al. (2020)	2	First line or pretreated with 1+ lines of therapy	Pembrolizumab + RT	8	0%	mPFS 1.4 mo mOS 2.9 mo	CBR 0%	12.5% G3 AE	PD-L1, TIL	NCT03366844	[46]
KBCRN-B002 (2023) Cohort A	1b/2	Pretreated with ≤2 lines of ET and ≤1 line of CT	Nivolumab + RT + ET of the physician’s choice	18	11%	mPFS 4.1 mo	DCR 39%	0% G3/4 AEs	PD-L1	NCT03430479	[47]
KBCRN-B002 (2023) Cohort B	1b/2	Pretreated with ≥2 lines of CT (must include taxane or anthracycline)	Nivolumab + RT	8	0%	mPFS 2.0 mo	DCR 0%	9% G3/4 AEs (includes TNBC pts, N = 10) *	PD-L1	NCT03430479	[47]

mo = months; DCR = disease control rate; NR = not reported; NA = not applicable; mDOR = median duration of response; CT = chemotherapy; ET = endocrine therapy; RT = Radiation therapy. * = Safety cohort includes TNBC (and HER2+) patients in the study.

## Data Availability

No new data were created or analyzed in this study. Data sharing is not applicable to this article.

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
