# Peer review of "Immune Checkpoint Inhibitor Therapy in Hormone Receptor-Positive Breast Cancer"

_ijms, 2025, doi:10.3390/ijms262412171_

Round 1

Reviewer 1 Report

Comments and Suggestions for Authors

The presented review is devoted to the summarization of the usage of immune checkpoint inhibitors in the therapy of hormone-responsive subtypes of breast cancer. The authors discussed the outcomes of immunotherapy in hormone-dependent breast cancer both in monotherapy and in combination with several classes of chemotherapeutics/targeted therapeutics. The review seems to be relevant to the scope of the journal, and the amount of the reviewed data is sufficient for the publication. However, there several issues to be addressed:

1.  Recent publication (O'Farrell J, Lapp C, Kuznia H, Afzal MZ. The Role of Immunotherapy and Immune Modulators in Hormone-Positive Breast Cancer: Implications for Localized and Metastatic Disease. J Clin Med. 2025 Jun 17;14(12):4322. doi: 10.3390/jcm14124322. PMID: 40566067; PMCID: PMC12194621, which could be accessed in PubMed web-resource) is devoted to the same topic as the presented review. Therefore, the topic is not new and the level of innovation is insufficient.

2.  The extensive English editing should be performed. Typos, mistakes, repeats should be eliminated. The descriptions of multiple clinical trials should be more concise.

3.  Abbreviation list is missing.

4.  The preclinical studies of the experimental approaches of immunotherapy (novel immune therapies in combination with novel targeted molecules) should be described separately and should be summarized in a Table as in case of clinical trials.

5.  The section 4. Biomarkers Predicting Response to ICI in HR+/HER2-EBC should be also summarized in a Table with the description of the potential prognostic value.

6.  The Conclusions should be modified to be more specific. Practical application of the reviewed data should be briefly stated.

Comments on the Quality of English Language

The extensive English editing should be performed. Typos, mistakes, repeats should be eliminated. The descriptions of multiple clinical trials should be more concise.

Author Response

Comment 1: Recent publication (O'Farrell J, Lapp C, Kuznia H, Afzal MZ. The Role of Immunotherapy and Immune Modulators in Hormone-Positive Breast Cancer: Implications for Localized and Metastatic Disease. J Clin Med. 2025 Jun 17;14(12):4322. doi: 10.3390/jcm14124322. PMID: 40566067; PMCID: PMC12194621, which could be accessed in PubMed web-resource) is devoted to the same topic as the presented review. Therefore, the topic is not new and the level of innovation is insufficient.

Response 1: We appreciate the reviewer’s comments and acknowledge that there rising interest of better understanding of immune checkpoint inhibitors in breast cancer, particularly for hormone receptor-positive, HER2-negative disease. The O’Farrell paper provided different perspectives with a focus on endocrine resistance and approaches to overcoming endocrine resistance, including immunotherapy. While our current review paper provided a comprehensive systematic review of the development of immunotherapy in HR+ breast cancer, with a broader coverage of novel immunotherapy combinations.

Comment 2: The extensive English editing should be performed. Typos, mistakes, repeats should be eliminated. The descriptions of multiple clinical trials should be more concise.

Response 2: We thank the reviewer for this comment. We have done extensive editing of the current review and ensured accuracy in grammar and structure. From the clinical perspective, the details of each clinical trial are necessary in providing more comprehensive information for the audience, especially for the clinicians. Hence, we prefer to keep the current details.

Comment 3: Abbreviation list is missing.

Response 3: We thank the reviewer for this comment. An abbreviation list has been added.

Comment 4: The preclinical studies of the experimental approaches of immunotherapy (novel immune therapies in combination with novel targeted molecules) should be described separately and should be summarized in a Table as in case of clinical trials.

Response 4: We thank the reviewer for this comment. Due to the limited preclinical data in the literature, we elected to use descriptions instead of a table for the preclinical data we included.

Comment 5: The section 4. Biomarkers Predicting Response to ICI in HR+/HER2-EBC should be also summarized in a Table with the description of the potential prognostic value.

Response 5: We thank the reviewer for this comment. Currently, the most comprehensive biomarkers predicting ICI benefit often combine clinical, pathological, and genomic biomarkers, and we believe that this integrated approach cannot be accurately described through a summary table for individual biomarkers.

Comment 6: The Conclusions should be modified to be more specific. Practical application of the reviewed data should be briefly stated.

Response 6: We thank the reviewer for their comment. We have modified our conclusion to highlight practical implications.

Reviewer 2 Report

Comments and Suggestions for Authors

The manuscript titled "Immunotherapy in Hormone Receptor-Positive Breast Cancer" provides a well-organized and comprehensive overview of clinical trials evaluating immune checkpoint inhibitors in HR+/HER2- EBC. The subject matter is clinically relevant and may offer valuable insight for therapeutic decision-making in this patient population. With the minor revisions listed below sufficiently addressed, the manuscript could be suitable for publication.

  1. The manuscript presents a large number of clinical trial examples. To further enhance clarity and readability, I suggest summarizing the respective strengths and limitations of each therapeutic approach. Additionally, including a comparative table would help visually highlight key similarities and differences across studies.
  2. Figures and tables are not integrated into the main text. Please incorporate the corresponding figure and table references in appropriate locations.
  3. In Page 13, Section “4. Biomarkers Predicting Response to ICI in HR+/HER2- EBC,” it would be beneficial to categorize biomarkers based on their type and elaborate briefly on their clinical implications and predictive value for immunotherapy response.
  4. Please carefully proofread the manuscript for terminology accuracy. In several instances, pCR is mistakenly written as PCR.
  5. Finally, I recommend expanding the discussion to include potential strategies for converting immunologically “cold” HR+/HER2- tumors into “hot” tumors, which may enhance responsiveness to immune checkpoint inhibitors and improve the efficacy of immunotherapy.

Author Response

Comment 1: The manuscript presents a large number of clinical trial examples. To further enhance clarity and readability, I suggest summarizing the respective strengths and limitations of each therapeutic approach. Additionally, including a comparative table would help visually highlight key similarities and differences across studies.

Response 1: We appreciate the reviewer's comment. We have summarized the clinical trials discussed in Tables 1 and 2. On page 6, we highlighted the limitations of these earlier trials due to small cohort sizes and lack of control.

Comment 2: Figures and tables are not integrated into the main text. Please incorporate the corresponding figure and table references in appropriate locations.

Response 2: We thank the reviewer for this comment. We have incorporated corresponding figures and tables as suggested.

Comment 3: In Page 13, Section “4. Biomarkers Predicting Response to ICI in HR+/HER2- EBC,” it would be beneficial to categorize biomarkers based on their type and elaborate briefly on their clinical implications and predictive value for immunotherapy response.

Response 3: We appreciate the reviewer’s insight. Currently, the most comprehensive biomarkers predicting ICI benefit often combines clinical, pathological, and genomic biomarkers. Therefore, this integrated approach provides probably the best predictive value. None of the individual biomarkers has been shown to provide sufficient predictive power.

Comment 4: Please carefully proofread the manuscript for terminology accuracy. In several instances, pCR is mistakenly written as PCR.

Response 4: We thank the reviewer for this comment. Corrections have been made.

Comment 5: Finally, I recommend expanding the discussion to include potential strategies for converting immunologically “cold” HR+/HER2- tumors into “hot” tumors, which may enhance responsiveness to immune checkpoint inhibitors and improve the efficacy of immunotherapy.

Response 5: We appreciate the reviewer's comment. We have discussed converting immune cold tumors into immune hot tumors through combination therapies with ICI. We have previously discussed non-ICI therapies in our paper mentioned at the end of the Introduction; therefore, those were not included in this paper.

Reviewer 3 Report

Comments and Suggestions for Authors

Review Comments

This review article presents an overview of immunotherapies currently under clinical development and updated key results from clinical trials with a focus on HR+/HER2- breast cancer (BC), which may arouse the attention and interest from the researchers in the relevant field. Nevertheless, there are also several problems exist in this manuscript, which should be carefully revised before published in International Journal of Molecular Sciences.

  1. The content of the summarization of the whole review article in Introduction section should be further enriched.
  2. Special symbols should be avoided in section headings (e.g., the “±” in the heading of Section 2.1), which can be replaced by proper English words.
  3. As key immune cells, CD4+ T cells and myeloid-derived suppressor cells (MDSCs) should also be reflected in Fig. 1.
  4. The citation of Fig. 1 in the main text is absent.
  5. A figure or table relevant to Section 2.4 should be supplemented.
  6. The numerical citation form of the literatures in Table 1 and 2 are different from the ones in the main text.
  7. The format of the Reference section should be ulteriorly improved.

Author Response

Comment 1: The content of the summarization of the whole review article in Introduction section should be further enriched.

Response 1: We thank the reviewer for the comment. We have enriched our summary in the Introduction.

Comment 2: Special symbols should be avoided in section headings (e.g., the “±” in the heading of Section 2.1), which can be replaced by proper English words.

Response 2: We thank the reviewer for the comment. We have made changes to our titles.

Comment 3: As key immune cells, CD4+ T cells and myeloid-derived suppressor cells (MDSCs) should also be reflected in Fig. 1.

Response 3: We thank the reviewer for the comment. We have added both CD4+ T cells and MDSCs to our Figure 1.

Comment 4:The citation of Fig. 1 in the main text is absent.

Response 4: We thank the reviewer for the comment. We have added context before section 2.1 to integrate Figure 1 into the main text.

Comment 5: A figure or table relevant to Section 2.4 should be supplemented.

Response 5: We thank the reviewer for the comment. The summary of the clinical trials discussed in section 2.4 can be found page 8 in Table 1.

Comment 6: The numerical citation form of the literatures in Table 1 and 2 are different from the ones in the main text.

Response 6: We thank the reviewer for the comment. We have updated the numerical citations in the Tables.

Comment 7: The format of the Reference section should be ulteriorly improved.

Response 7: We thank the reviewer for the comment. We have updated the references style.

Round 2

Reviewer 1 Report

Comments and Suggestions for Authors

Authors stated that they prepared systematic review but there are separate requirements for systematiс reviews (for example, PRISMA). If there is a systematic review, authors should prepare the manuscript according PRISMA guidelines

Reviewer 3 Report

Comments and Suggestions for Authors

This manuscript is suitable to be published in IJMS after revision.